# Fast and Interactive Positioning of Proteins within Membranes

André Lanrezac [1], Benoist Laurent [1], Hubert Santuz [1], Nicolas Férey [1,2] and Marc Baaden [1,*]

1. Laboratoire de Biochimie Théorique, CNRS, Université Paris Cité, 13 Rue Pierre et Marie Curie, F-75005 Paris, France
2. Laboratoire Interdisciplinaire des Sciences du Numérique, CNRS, Université Paris-Saclay, 91405 Orsay, France
* Correspondence: baaden@smplinux.de; Tel.: +33-1-58-41-51-76

**Abstract:** (1) Background: We developed an algorithm to perform interactive molecular simulations (IMS) of protein alignment in membranes, allowing on-the-fly monitoring and manipulation of such molecular systems at various scales. (2) Methods: UnityMol, an advanced molecular visualization software; MDDriver, a socket for data communication; and BioSpring, a Spring network simulation engine, were extended to perform IMS. These components are designed to easily communicate with each other, adapt to other molecular simulation software, and provide a development framework for adding new interaction models to simulate biological phenomena such as protein alignment in the membrane at a fast enough rate for real-time experiments. (3) Results: We describe in detail the integration of an implicit membrane model for Integral Membrane Protein And Lipid Association (IMPALA) into our IMS framework. Our implementation can cover multiple levels of representation, and the degrees of freedom can be tuned to optimize the experience. We explain the validation of this model in an interactive and exhaustive search mode. (4) Conclusions: Protein positioning in model membranes can now be performed interactively in real time.

**Keywords:** interactive molecular simulation; implicit membrane; lipid bilayer insertion; protein orientation; coarse-grained representation

## 1. Introduction

Proteins are an essential component of all cells and perform a variety of functions. An important role of proteins is that they act as functional and structural elements of cell membranes, the fluid barrier that separates the cell from its environment [1]. The position of the protein within the membrane is a critical element in understanding many processes and can be predicted computationally, since direct experimental measurement is not routinely possible [2]. The incorporation into the membrane can affect the stability, folding, and activity of the protein, as well as its interactions with other molecules such as ligands, substrates, or surrounding macromolecules. To take a single biological example, for membrane fusion, the incorporation of transmembrane domains into the membranes to be fused is crucial [3–5]. Several databases have been developed to collect information on membrane incorporation and alignment, such as [2,6,7]. In certain cases, direct linkage of computational predictions with experiments is possible, for example, with data from electronic paramagnetic resonance experiments [8]. Computational algorithms range from simple implicit membrane representations [9] to computationally intensive molecular dynamics simulations in fully hydrated lipid bilayers [10,11] and are not interactive. Interactive molecular simulations (IMS), as recently revisited in [12], offer a different approach: the scientist performs the manipulations interactively, providing intuitive insight into the process. Biologically and functionally relevant structural information can be obtained through IMS. The purpose of this work is to develop an interactive approach for protein positioning in membranes. By extending a robust bioinformatics method into an interactive method, non-experts can effortlessly perform and analyze such calculations, offering a valuable and novel tool to

complement experimentally oriented studies. This use of interactive simulations is particularly interesting in the context of integrative structural biology [13], where inaccuracies in computational predictions are corrected by the experimental data guiding the refinement until a match between the computational model and the experimental characterization is achieved.

The positioning of a protein in the membrane can be defined by several geometric measures, the most important of which are the insertion depth and angle. The insertion depth is the distance of the center of mass of the protein from the plane of the center of the membrane, measured in the direction perpendicular to this plane. It is common to refer to this direction as the *z*-axis of the coordinate system. To calculate the insertion angle, we define a local axis passing through two residues of the protein and calculate its angle with respect to the membrane normal, e.g., the *z*-axis.

The research field of computational assessment of membrane protein positioning and incorporation into lipid bilayers has reached an advanced level of maturity, but interactive approaches have been largely neglected. Here, we build on the method of Integral Membrane Protein and Lipid Association (IMPALA) developed by Brasseur's team, which studied the interactions of different proteins with the implicit membrane representation. We chose this method because it is simple and computationally efficient, making it well suited for real-time interactive experiments and subsequent incorporation into an integrative modeling workflow. Molecular modeling studies using IMPALA, for example, aimed to calculate the conformations of a nisin compound in the membrane, confirming experimental results [14], and helped characterize new synthetic cell-penetrating peptides [15] or known peptides [16]. The orientation and insertion depth of integral membrane proteins were also studied in [17]. The main parameter of interest is the insertion angle, which provides information about the configuration that a protein adopts in the membrane according to its structural hydrophobicity. Indeed, the amphipathic protein insertion performed and presented in the original IMPALA article aimed to confirm some experimental results [18], especially in NMR, where the orientation of the protein is the result of a complex interpretation of the parameters of the experimental procedure [19]. The interest to experimentally study the orientation of membrane proteins was formulated in this work as follows: "These structural arrangements established on the basis of spectroscopic measurements are now beginning to influence the descriptions of the mechanisms of actions of these peptides in membrane bilayers".

Our aim was to develop a computationally efficient, interactive algorithm that is scalable to large membrane proteins and provides immediate visual feedback on orientation and incorporation. We describe the key factors, implementation, and evaluation of the method. The algorithm should be flexible enough to accommodate multiple scales of protein representation and provide live visualization of protein position and orientation with respect to the membrane plane. The main conclusions from our studies are twofold. First, we present a generalizable approach to make molecular computations interactive. Second, we demonstrate the complementarity of the interactive approach to membrane positioning compared to previous methods.

## 2. Materials and Methods

We first describe the original reference method by Brasseur's team on which our work is based and how membrane insertion is conceptualized by several energy terms describing the water–bilayer interface, a hydrophobic restraint term, and a contribution to the perturbation of lipid molecules. Next, we describe the software components required for an interactive approach and how the reference method for membrane insertion can be integrated into the interactive context. Given the real-time constraints, we then describe the required optimizations and extensions to the implementation. Since we want to make the method interactive for a wide range of systems, we implement performance optimization by using a rigid body method to reduce the number of degrees of freedom and extend the molecular representations to multiple levels of representation, from detailed all-atom to

simplified coarse-grained models. We enable the possibility of conformational changes by making the models flexible on demand using an elastic network representation, requiring real-time recalculation of their exposed surface. The latter is necessary to assess atomic interactions that depend on the solvent accessibility or exposure to the lipid phase, such as hydrophobicity or insertion using IMPALA. To evaluate and validate our implementation, we have provided the capability to perform automated parameter scanning to systematically explore the insertion depth and angular alignment parameters. We provide hardware considerations and performance measurements. To perform the interactive experiments, it is necessary to monitor the process, design interaction metaphors, and perform visual analysis. Most of our calculations are focused on one test system extensively described in the literature but also by Brasseur's works, the OmpA outer membrane protein, a beta barrel, for which we use PDB-ID 1BXW [20].

### 2.1. Original Reference Method for Integral Membrane Protein and Lipid Association (IMPALA)

The empirical approach of Integral Membrane Protein and Lipid Association (IMPALA) was introduced in [17]. It is a method for studying the association of molecules with an implicit membrane represented by an analytical function C(z) that depends on a single variable **z** to describe the lipid–water interface along the membrane normal of a lipid bilayer. A detailed description of this method can be found in [18]. Proteins are represented by seven atom types, with two types for carbon: double-bonded $C_{sp2}$ including aromatic cycle carbons, or single-bonded $C_{sp3}$, O, S, N, non-charged hydrogen bonded to C and charged H.

#### 2.1.1. Water–Bilayer Interface Description

Figure 1 gives an overview of the form of the mathematical equation for C(z):

$$C(z) = 0.5 - (1 + \exp(\alpha(|z| - z_0)))^{-1}, \tag{1}$$

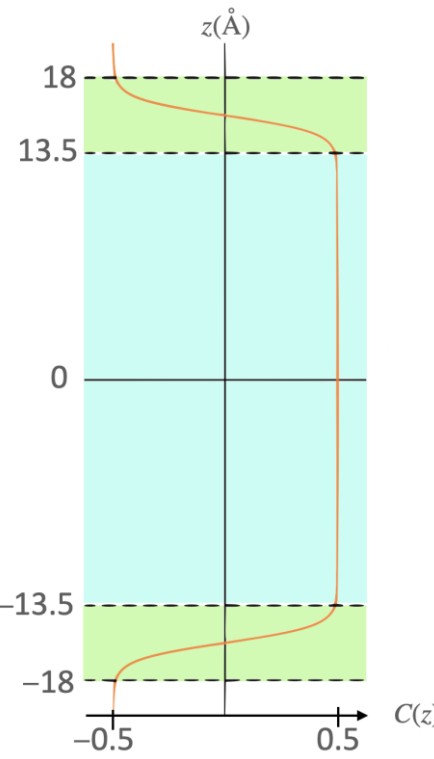

**Figure 1.** Description of the water–membrane interface by the C(z) function, which varies from −0.5 to +0.5, corresponding to the compartments outside of the bilayer and inside its hydrophobic core, respectively. The hydrophobic core is shown in cyan; the lipid headgroup domains are shown in green.

The variable **z** in ångström is the membrane penetration depth of a single molecular particle (typically an atom). The function behaves like a switch between two values ($\pm 0.5$) varying along the $z$-axis, which is perpendicular to the membrane plane, to assign a value to each particle that describes its membrane environment. This is an empirical equation because its parameters are chosen to correspond to the structural properties of a biological membrane: $|z| \in [13.5; 18]$ Å (in green) represents the two transition zones that qualitatively describe the presence of polar head groups between the hydrophilic phase located at $|z| > 18$ Å and the hydrophobic membrane core for values of $|z| < 13.5$ Å (in blue). With this simple description of the membrane environment of a single particle, generalization to the entire structure of a macromolecule simply yields an insertion value that depends on its orientation and position, defined by two rotation angles and one translation distance along z in the membrane. This function alone is not sufficient to describe the interaction of the molecule with the membrane.

### 2.1.2. Hydrophobic Restraint Term

Membrane insertion is described by empirical data characterizing the physical property of hydrophobicity of the molecule. This property drives insertion by pushing the hydrophobic regions into the membrane and, conversely, driving the hydrophilic regions outward. Each atom type is associated with a transfer energy term in kJ·mol$^{-1}$·Å$^{-2}$. This energy term represents the energy that must be supplied to an atom of a given type to be transferred from a hydrophilic to a hydrophobic environment, given its accessible surface area in Å$^2$. An accurate description of the hydrophobic effect takes into account that the solvent interacts only with accessible atoms and not with those buried in the molecular structure. The hydrophobic restraint term is consistent with two experimental facts: the hydrophobic effect is related to the nature of the solute and to the surface area of the solute in contact with the solvent. The transfer energies per atom were derived from the assumption that each exposed surface region contributes equally to the total transfer energy of the residue, which was taken from the work of Fauchère and Pliska [21].

The restriction term is calculated as

$$E_{int} = -\sum_{i=1}^{N} S_{(i)} \, E_{tr(i)} \, C_{(zi)}, \tag{2}$$

and acts only at the interface. Figure 2 shows this restriction term applied to a sp3 carbon type with an accessible surface area of 56.78 Å$^2$. The arrows at each interface show the derived forces that would push the carbon into the hydrophobic core of the membrane. The central curve in black on the $z$-axis shows on the $y$-axis the amplitude of the force acting on that atom, which is negative on the left side and positive on the right side of the $z$-axis.

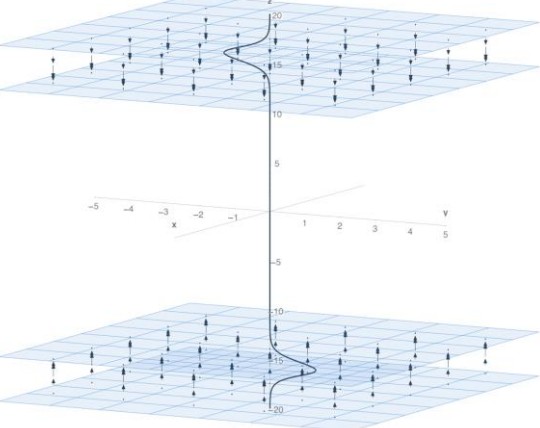

**Figure 2.** Illustration of the hydrophobic constraint term and the direction of the resulting forces in the headgroup regions of the lipid bilayer.

### 2.1.3. Lipid Molecule Perturbation Restraint

An energy penalty term is added to account for the disturbance caused in the lipid bilayer by the insertion. This term reflects the fact that the protein disrupts the lipid bilayer when it inserts and therefore tends to minimize its exposed surface with the membrane. The nature of the accessible surface, whether hydrophobic or hydrophilic, is not taken into account. Therefore, this term always represents an additional penalty for atoms with nonzero accessible surface area that would have entered the membrane and disrupted lipid interactions.

$$E_{lip} = a_{lip} \sum_{i=1}^{N} S_{(i)} C_{(zi)}, \tag{3}$$

with the empirical factor $a_{lip} = -0.018$ kcal mol$^{-1}$ Å$^{-2}$.

### 2.1.4. Full IMPALA Potential Energy Term

The resulting potential energy term IMPALA, which represents the total energy balance, is the sum of the restraints of the hydrophobic transfer and lipid bilayer perturbation. The corresponding energy and force expressions are as follows:

$$E_{IMP} = E_{int} + E_{lip}, \tag{4}$$

$$F_{IMP} = d/dz (E_{int} + E_{lip}). \tag{5}$$

These terms contribute to the overall description of the potential energies and forces of the molecular system driving the simulation.

### 2.2. *Software Components for Integration of IMPALA in an Interactive Setup*

Interactive molecular simulations allow the control and visualization of a molecular simulation, such as an ongoing membrane insertion. To this end, we developed a generic library called MDDriver that facilitates the implementation of such interactive simulations [22]. MDDriver allows the easy creation of a network socket between a molecular visualization user interface and a physics-based simulation engine. We use our own molecular visualization tool, UnityMol [23], to drive the dynamic behavior of the molecule in our IMPALA implementation. The relationship between these three software components is shown schematically in Figure 3 and uses standardized protocol and data structures for information exchange. More details on such interactive simulations can be found in [12].

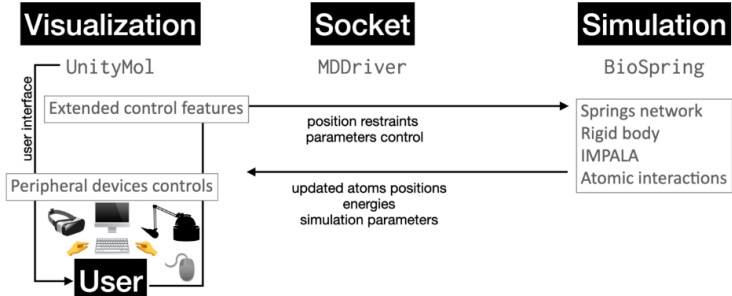

**Figure 3.** Interactive molecular simulations rely on three pillars: visualization, a network socket layer, and a simulation engine. Here, we show the three software components used in the present implementation and illustrate the data flow and task distribution.

The visualization part uses the Unity game engine for development, whose C# language allows for modularity. It is possible to quickly test new features in a dedicated and proven framework for user experience, device management, user interface, etc. The IMD protocol is implemented natively in UnityMol and communicates directly with the MD-Driver socket for simulation data exchange. Likewise, it is possible to extend the exchange to new data types by adding interaction features, as described later. The implementation of the IMPALA algorithm was built on our interactive simulation engine BioSpring [24].

BioSpring, similar to molecular dynamics simulation engines, uses additive energy and force terms to represent the various contributions to a given model representation. Thus, the main task in implementing a new computational method such as IMPALA is to compute the forces and energies during the main simulation loop and add them to the other forces and energies, at least those generated by the user in an interactive experiment, and possibly to other force field or elastic network terms if, for example, flexibility is to be considered. The steps required to implement IMPALA are shown as pseudo-code in Figure 4.

```
// initiation

configure_FreeSASA() // set FreeSASA parameters

// main loop

while is_biospring_running do

    all_sasa <- call_FreeSASA()

    for i <- 0 to particles_length:

        particles[i].sasa <- all_sasa[i]

    end for

    restraint <- 0

    for all p ∈ particles do

        restraint <- restraint + compute_IMPALA_restraint(p.position, p.sasa, p.tr)

        p.force <- p.force + [0, 0, 1] * compute_IMPALA_force(p.position, p.sasa, p.tr)

    end for

    if spring_enabled then // spring network algorithm

        // particles forces integration

    else if rigidbody_enabled then // rigid body algorithm

        // particles forces integration

    end if

end while
```

**Figure 4.** Pseudo-code for adjustments to the main BioSpring calculation loop to accommodate the interactive method for calculating membrane insertion.

Before entering the main calculation loop, you may need to set some parameters. Within the loop, the surface area accessible to the solvent is first calculated for all particles, as it is entered into both Equations (2) and (3). Next, the IMPALA energy is calculated and

added to the existing restraints, such as user-generated forces. Depending on whether you selected the option to enable molecular flexibility or to use a static model, either the spring network routines or the rigid body routines are then called to update the positions of the molecule under investigation. The energy calculations of IMPALA can easily be replaced by any alternative calculation of membrane insertion energetics that is desired. In future developments, one could envision a library of different methods from the literature from which the user could select on the fly the algorithm best suited for a particular task.

### 2.3. Optimization and Extension of the Implementation

Interactive simulations are inherently subject to performance constraints, as they need to run and update fast enough for the user experience to be smooth and interactive in real time. We therefore explored several options to improve performance, such as reducing the degrees of freedom of the molecular representation by coarsening it, optimizing the parameters to construct the elastic network inspired by considerations reported in the literature [25], or removing flexibility in a rigid body approach. Improvements were also useful to have either flexible molecules where the surface changes in real time due to conformational changes, or the ability to assemble multiple molecules in the membrane where the surface evolves because some parts are buried by aggregation. To generate data that are comprehensive and not subject to user interaction for validation purposes, we also found it useful to implement a function that systematically searches a given parameter space. In terms of performance optimization, we tested and compared different hardware configurations that we had easy access to in order to optimize the usability and fluidity of interaction.

### 2.3.1. Extension of IMPALA to Coarse-Grained Representations

The original implementation of IMPALA was based on an all-atom representation. Nowadays, coarse-grained representations are commonly used to model membrane systems and allow the management of large and complex molecular assemblies. To limit the computational costs associated with an all-atom representation, we started to incorporate the coarse-grained representation of Martini version 3 [26] into our implementation. The BioSpring program has a general procedure for particle reduction. It is necessary to generate a set of instructions for this procedure so that it can represent the grouping of one or more atoms in Martini 3 grains. In particular, two input files are needed to perform a reduction procedure. The first contains the instructions for grouping the atoms into so-called large grain particles, and the second contains the parameters associated with each type of grain (electric charge, radius, Lennard–Jones energy parameters, mass and transfer energy for IMPALA). The input files are used to translate a PDB structure into CDL format. CDL stands for Common Data Language, a type of human-readable dataset from the netCDF library (or its binary equivalent), and is required by the BioSpring engine to set up a simulation. Most of the work required to integrate Martini 3 into BioSpring was to implement the code that formalizes the nomenclature of Martini 3 grains and their parameters to generate input files that can be used to reduce any molecular system into Martini 3 grains. In particular, in the context of membrane insertion, a new method was needed to calculate the contribution of the transfer energy used for IMPALA. The required information can be found in reference [26] (particularly in its Supplementary Tables S17–S19 on pages 15–17). The free energies of transfer were extracted from this work and divided by the calculated theoretical surface area of each grain type (normal, small, and tiny) to give an equivalent for the energy of transfer per surface unit of individual atoms in kJ mol$^{-1}$ Å$^{-2}$, as shown in Table 1 of [18]. A future extension would be to specify additional distance constraints between CG particles, typically the internal distances between grains of the same amino acid residue formalized in Martini. The current implementation specifies the position of the CG particle on the center of mass of its particle group. At first glance, the implementation of such a geometric calculation is not trivial and will require some development time. Therefore, the current method provides an alternative for taking these spacing constraints into account

by defining certain equilibrium distances between springs. To ensure that the inter-grain spacing meets the Martini specifications, a relaxation step is required prior to simulation. This implementation, of course, is only a temporary solution.

**Table 1.** Surface area calculation methods comparison.

|  | 1PPT | 4PTI | 2PCY | 2RHE | 5NLL * | 2UTG |
|---|---|---|---|---|---|---|
| $A_{num}(\text{Å}^2)$ | 3327.5 | 3970.9 | 4954.9 | 6246.7 | 6939.9 | 7192.8 |
| %$_{NACCESS}$ | 0.17 | 0.02 | 0.26 | −0.06 | 0.27 | 0.61 |
| %$_{FreeSASA}$ | 0.10 | −0.12 | −0.03 | 0.10 | 0.38 | 0.59 |

* supersedes 3FXN.

### 2.3.2. Adding a Rigid Body Positioning Algorithm to BioSpring

For many applications, the internal conformational flexibility of the protein to be incorporated is not required. Therefore, the computations associated with these degrees of freedom that would be required, for example, in an elastic network representation are unnecessary. We therefore implemented a rigid body model in BioSpring to avoid such unnecessary calculations. A customized rigid body dynamics algorithm is necessary to preserve the structure of the protein and thus keep all internal relative positions constant. Such a procedure allows a drastic reduction in computational cost by reducing the number of degrees of freedom. The minimal implementation required to perform our experiments is to allow a rigid structure whose particles can be controlled individually and whose global translational and rotational motions are computed according to all the forces acting individually on the particles, including the forces derived from the equations of IMPALA. The rotational motions of the rigid body are always performed around its center of mass. The state variables of the rigid body are its position—more precisely, the position of the center of mass—its translational velocity, and its angular velocity defined by the instantaneous axis of rotation, whose norm and direction indicate the amplitude and direction of the direction of rotation. The angular velocity is calculated directly from the torque acting on the entire protein. During the interactive simulation, the position of the center of mass, the angular velocity, and the torque may be displayed in the form of lines updated according to the data received from BioSpring.

Our implementation is minimal and lacks some features typically found in other rigid body implementations, as they are not currently needed for our purposes. The functions needed to calculate the protein inertia matrix are implemented, but are not called in the rigid body force integration routine. The reason for this is that the complexity of the implementation is high, and for insertion into implicit membranes, it is quite sufficient to consider the rigid protein as an object with uniformly distributed mass. We have also simplified the rotation calculations, which are not performed with quaternion multiplication as originally implemented. At each iteration, we sum all the forces acting on the particles and derive the torque, angular velocity, and translational velocity of the entire protein. Both velocities are then used to calculate the velocities of the individual particles as the sum of the translational velocity and the local angular velocity with respect to the center of mass. Thus, the velocities are solved explicitly in a full timestep. So far, we have not gone the route of nested timesteps as in the Verlet or Leapfrog algorithms, which are widely used (and required) for other types of biomolecular simulations.

### 2.3.3. Real-Time Calculation of the Protein's Accessible Surface Area

As can be seen in the pseudo-code of our implementation, we called the FreeSASA library to dynamically evaluate the accessible surface, which is important if the molecule is to be represented flexibly, for example, with an elastic network representation. We validated our implementation on some globular proteins used as models in the comparative study presenting the Analytical Surface Computation (ASC) program [27]. In particular, they compared their analytical method in terms of difference in accuracy with the "numerical" method, a variant of the Shrake–Rupley algorithm. We use the results of the latter as

comparison results referenced as $A_{num}$ in Table 1. We also consider values obtained with the double cubic lattice method (DCLM), a variant of the Shrake–Rupley method, and its implementation called NSC (Numerical Surface Computation) [28]. Here, we specifically compare the results obtained with two widely used ASA calculation programs, NACCESS [29] and FreeSASA [30]. An inherent difficulty is that not all details necessary for reproducibility are included in the original publications of Brasseur's team. We have conducted thorough research to develop a more accurate idea of the atomic radii used in Brasseur's experiments with NSC. The NSC reference article seems to indicate that the van der Waals atomic radii are the reference radii. The Define Secondary Structure Program (DSSP) article [31], which uses NSC, specifically mentions some of the radii used: 1.40 Å for O, 1.65 Å for N, 1.87 Å for Cα, 1.76 Å for C and CO in the backbone, and 1.80 Å for all sidechain atoms. These radii correspond to the radii used by NACCESS using the standard van der Waals configuration file for the comparisons presented in this section (excluding sidechain atoms at 1.80 Å and excluding N_AMN of Lys at 1.50 Å). We therefore use these radii to perform our surface area calculations with FreeSASA, which we use for our IMPALA experiments in BioSpring. The parameter values used can be found in the deposited data. In addition, the experiments performed by Brasseur's team suggest the addition of hydrogen atoms, which we therefore also added in our experiments using the pdb2pqr tool. We chose a hydrogen atomic radius of 1.0, which is the default value in the NACCESS code (in comparison, the corresponding value is 1.1 in FreeSASA). We try to use the parameter set that we assume is closest to Brasseur's computational conditions. This is discussed in more detail in Appendix A. We are aware that the choice of atomic radii is crucial for the calculation of the surfaces and thus of the IMPALA energy term. Our set of parameters does not exactly match those used in [17], but our results shown in Table 1 only minimally differ, by up to 0.6% in calculated surface area. However, it should be noted that we are not completely sure about the equations used in [17] for the IMPALA clamping energy. Therefore, we use the equations presented in the reference article of the IMPALA method [18].

### 2.3.4. Automatic Parameter Scanning and Simple Monte Carlo Approach for Protein Insertion

We implemented a function that samples all positions that the protein can occupy in whole or in part in the membrane with discrete values offset by 1 Å or 1°. Such sampling is feasible for simulations with a personal computer in a reasonable amount of time because there are only three degrees of freedom that can be varied for a system of size N. These are insertion depth, insertion angle, and rotation along the axis passing through the insertion vector, also referred to as roll angle. Such scans are useful to evaluate and validate our implementation and to extend the interactive exploration by the user to compare both approaches. Typically, this parameter scanning means sampling 360 steps for roll angle, 180 steps for insertion angle, and at least 36 steps for insertion depth, which is equivalent to 2.3 million energy evaluations per full scan.

As described in Section 4 and Appendix A, in addition to the parameter scanning approach, we implemented a simple Monte Carlo sampling algorithm for comparison to mimic the original calculations. To sample under similar conditions, we use one translational and one rotational move of the membrane protein to be inserted.

### 2.3.5. Hardware Considerations and Performance Measurements

Since BioSpring can run on multiple threads using OpenMP, we benchmarked it to evaluate its performance. First, we used 4 different computers with different hardware: an Intel® Xeon® E5-2630 v4 @ 2.20GHz (released in 2016)-10 cores, 20 threads (sablons); an Intel® Xeon® E5-1620 v2 @ 3.7GHz (released in 2013)-4 cores, 8 threads (han); an Intel® Core™ i5-10500 @ 3.1GHz (released in 2020)-6 cores, 12 threads (hal); and an Intel® Core™ i7-1185H @ 2.50GHz (released in 2021)-8 cores, 16 threads (dell). We measured the number of iterations per second of the BioSpring program as a function of the number of threads (Figure 5) on a system with 12,000 particles.

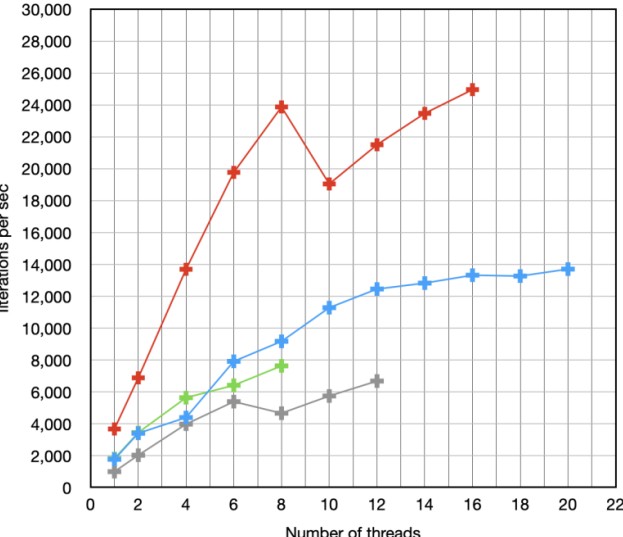

**Figure 5.** Performance benchmark on a structure with 12,000 particles. The colored crosses correspond to the workstations described in the text, with sablons, han, hal, and dell shown in blue, green, gray, and red, respectively.

The results show that the most recent CPU achieves the best performance, i.e., the highest number of iterations per second. The second result is that the frequency at which the CPU unit runs is less important than the number of threads that can be generated. The workstation sablons performs better than the workstation han thanks to the number of cores and despite its lower frequency. The workstation hal performs the worst of the 4 benchmark machines, which could be due to the type of CPU unit (i5 compared to Xeon and i7). Another observation is that the number of OpenMP threads should not exceed the number of CPU threads on the workstation, otherwise performance will drop.

We also compared our regular implementation of BioSpring to the rigid body implementation we developed on the workstation hal. We ran a simulation with 10,000 steps using IMPALA and measured the elapsed time. The results showed that using a rigid body approach increases the performance for the OmpA system by 54%.

### 2.4. Monitoring, Interaction Design and Visual Analysis

We have developed a number of tools and functions to monitor the insertion process. This visual analysis primarily requires a precise definition of key descriptors, which is described below. We added an interaction design to UnityMol and BioSpring to perform interactive insertion experiments, which we discuss next. Several visual analysis features have been implemented to inform the user during such an experiment.

#### 2.4.1. Definition and Measurement of Important Descriptors

Insertion angle. For the implementation, we apply the controlled simulation of the insertion angle parameter by defining an insertion vector based on two particles. These two particles are selected according to the alpha carbon atoms CA of the reference residue pairs described in [17] that define a reference secondary structure element—in our case, a beta strand. The insertion vector is different from the major axis of the protein (e.g., the central axis of a β-barrel), and so the insertion angle gives the tilt angle of the membrane protein strands for β-proteins with respect to the membrane surface, as explained in [17]. We create a reference axis that passes through the insertion vector. This reference axis allows us to freely simulate the degree of freedom of the roll rotation, which is not captured by the insertion angle alone. The roll value is not explicitly captured, but evolves according to the trajectory of the protein at a given insertion angle. The pseudo-code of the corresponding function "computeAngle" is given in Figure 6.

```
void InsertionVector::computeAngle()

{

    _insertionAngle = std::acos(-_insertionVector.getZ() / _insertionVector.norm());

    _insertionAngle *= 180.0 / M_PI;

    _insertionAngle -= 90.0;

}
```

**Figure 6.** Pseudo-code for computing the insertion angle.

In the implementation of the automatic scan, a rotation of 180° range is performed with a step of 1° along the axis perpendicular to the unit vector z and the insertion vector (vector product). This range corresponds to a variation of +90° to −90° as the latitude of the geographic coordinates.

Insertion depth. More precisely, the insertion depth is the value of the z-component of the center of mass (COM) of the protein, with the *z*-axis being perpendicular to the membrane plane. For proteins with detailed information about all atoms (including hydrogen atoms), the COM is the average of the positions of the atoms weighted by their mass. We calculated the initial and final z values as a function of the distance $d_{max}$, beyond the implicit membrane width spanning from +18 Å to −18 Å, that lies between the COM and the structurally most distant atom. Thus, the z value of the COM varies with a decrease of −1 Å at each step in a range of $\pm(18 + d_{max})$, where 18 is the z value in Å of the outer membrane boundary.

Roll angle. The roll rotation along the axis passing through the insertion vector varies from 0 to 360 degrees. We included the measurement of the roll angle in this study. It is a degree of freedom that was not explicitly mentioned in the studies of Brasseur's team, although it was necessarily simulated as a hidden variable in their Monte Carlo experiments. In fact, this variable seems to be important because there are several possible positions for a given insertion angle. It is therefore necessary that the sampling approach explicitly contains this roll rotation for all insertion angles. On the other hand, we do not keep all steps for each roll angle, but only the step for a particular angle that has the lowest energy among all roll angles, which is a form of dimensionality reduction. It is, of course, possible to find multiple local energy minima at different roll angles for the same insertion angle. In this case, exploratory research with interactive simulations is useful to discover the transition states between multiple local minima.

2.4.2. User Control and Interactive Exploration Tools

The IMD protocol offers the possibility of directly grabbing a particular atom or atoms and pulling on them as if they were attached to a spring. Such a manipulation simultaneously affects depth and orientation. The interactive procedure performed with BioSpring is designed so that the variable the user should primarily try to control interactively is the insertion depth, while the IMPALA constraints can freely vary the orientation of the protein in the membrane. Subsequent manual fine-tuning of the insertion angle is, of course, possible. Several tests were performed where the user can directly and simultaneously select both parameters (insertion angle and insertion depth). However, the interpretation of the trajectory resulting from such an experiment, which records the path that the protein travels in the membrane to reach a given orientation, is quite uncertain. This is true even when the user-imposed constraints are executed for only a short time, followed by relaxation periods. A Steered Molecular Dynamics (SMD)-inspired anchoring method that manually positions an anchor to which the atoms are attached has been developed but is not yet complete. Using the keyboard, the user can control the protein like in a video game to move it up or

down in the membrane and rotate it in all directions. A special button allows the user to shake the protein to overcome some energetic barriers in exploring the phase space.

2.4.3. Interactive Visualization and Analysis with UnityMol

Recent additions to the visual feedback are that the accessible surfaces are updated in real time by BioSpring and displayed as color variations of the protein atoms shown in UnityMol. Other interactive features managed through the UI, which is always in direct communication with BioSpring through MDDriver, are the display of the insertion vector, the implicit membrane, the computational lattices of some atomic interactions, or all kinds of multiplication factors of the simulation calculations. Such plots can be used to monitor the values of simulation status variables, such as insertion depth and angle, IMPALA energy, etc. The 2D (depth vs. angle) graph displayed in UnityMol is interactive and allows the mouse to be clicked on a user-selected location to steer the protein to the desired insertion angle and depth. It is possible to disable the angle control and retain only the depth control, allowing the system to position itself at the most favorable angles and freely apply the IMPALA constraints to vary the orientation of the protein in the membrane. In this way, it is possible to see the effects of varying one degree of freedom (depth) on the other parameters (angle) and vice versa.

Two additional cameras were added to the Unity scene at the x- and *y*-axis viewpoints in orthographic projection. The renderings of these cameras are directly visible in the main window and show the simulated protein penetrating the membrane from a fixed viewing angle. The images from these cameras can be saved with the exact angles and depths of insertion for analysis. Python code was developed to process these data and interactively verify the positions visited by the protein in the membrane at a given step using a linking and brushing approach [32].

**3. Results**

The integration of IMPALA into our BioSpring simulation engine has been extensively tested on a system previously studied in detail in the work of Basyn [17]. We use the OmpA porin, a well-studied beta-barrel membrane protein, to evaluate the results of the interactive IMPALA algorithm described in the Materials and Methods section. The insertion calculations originally published in the literature were performed using a Monte Carlo method, which we compared with our interactive method. Their experiments were performed with static structures, i.e., they inserted rigid body protein models of fixed conformation into the membrane environment. We therefore implemented a comparable method in our protocol. In this section, we describe in detail the studies that were performed to reproduce as closely as possible the original experimental conditions. The problem of reproducibility is also addressed and discussed.

*3.1. Comparison of IMPALA/BioSpring and Reference Data*

We were interested in validating our implementation with respect to the studies of Brasseur's team on the OmpA protein, so we used a rigid body representation. The rigid body dynamics were implemented in our BioSpring engine specifically for this study, as described in Section 2. For validation purposes, we used exhaustive sampling with discrete values offset by 1 Å or 1° for all positions that the protein can occupy either fully or partially in the membrane. First, we applied our protocol to determine the energies for each insertion depth and angle separately, as in the results in Figure 2 of [17]. For direct comparison, we extracted the data points from the original plots using the WebPlotDigitizer tool [33] and integrated them into the plots as blue points. The energy term IMP is defined in kcal/mol and is a restraint term rather than an actual energy function, since it does not correspond to the actual physical interactions [18]. The residues chosen to represent the insertion vector and thus measure the angle with respect to the membrane are Lys34 and Val45, which correspond to the reference strand (second β-strand) given in the original publication. However, there is an ambiguity because, in contradiction to this indication in the text of the

article, residues 31B–36B, i.e., residues 31 and 36 of chain B, are indicated in Table 2 of [17]. However, only K34 and V45 agree with the structural representation in Part A, Panel (F) of Figure 3 of [17], so we discarded the other pair. The particles chosen to represent this vector are the $\alpha$-carbons of these residues.

Figure 7 shows 9523 steps, where each successive step corresponds to a position where the insertion angle varies from 0° to 90° in steps of 1°. This range of positive insertion angles (between 0° and 90°) was chosen to allow consistent comparison with Brasseur's data and to measure the same range of insertion angles. As explained above, each step also corresponds to the minimum energy roll for a given insertion angle. The experiment was performed for OmpA with an insertion depth of about 53 Å to −53 Å, as we can see in Figure 7A. This range extends the insertion to the entire membrane and shows us in Figure 7C that the preferred insertion depth for positive insertion angles is about −5 Å +/− 5 Å, while the reference value given by Brasseur is −7 Å. These preliminary results for the insertion angles do not show clear preferred ranges for the angles, but some local stabilities around 20°, 35°, and 45°. For OmpA, Brasseur's reference angle is 50° to 55° with respect to the membrane surface. However, they point out in their paper that the literature assumes that all strands (of which the insertion vector is a representative) are tilted by about ∼45° with respect to the membrane surface, which is consistent with our results. We also recall that it remains unclear how exactly the roll degree of freedom was handled in the reference work. Since we find non-negligible differences in the insertion angles between the approach proposed here and the reference Monte Carlo calculations, further analysis is required. For a more detailed discussion, see Section 4. Appendix A contains additional experiments and references to alternative solutions. It is interesting to note that this automatic sampling procedure seems to reproduce some trajectory-like patterns that we will find in the interactive simulations in the next section.

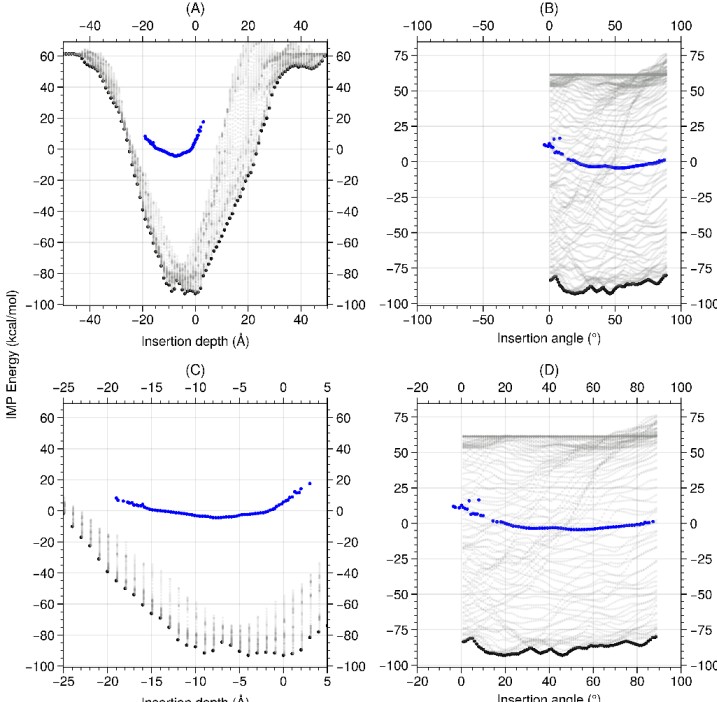

**Figure 7.** Automatic scanning of the membrane insertion of the OmpA membrane protein. The blue dots correspond to the energies of the reference Monte Carlo calculations, as indicated in the text. The results of our own calculations are represented by the gray and black dots. The black dots are the minimum energy values at a given insertion angle or depth, rounded to the nearest multiple of 0.5. The gray dots represent all other energy values that were visited by the scan. All plots share a common $Y$ axis on which the IMAPALA energy term is given in kcal/mol. (**A**,**B**) are plots showing the full range of values of the insertion depths and angles. (**C**,**D**) are enlargements of these plots with the same parameter windows as in the figure in the reference article.

### 3.2. First Interactive Experiments

We provide two videos of our interactive experiments: Supplementary Video S1 with a rigid body setup, and Supplementary Video S2 using a flexible model. To characterize these experiments, we first present figures corresponding to the ones shown in the previous paragraph, but now taken from interactive simulations. The gray dots give an idea of the non-relaxed positions explored during the simulation and thus depend on the user's control, but also on the protein's ability to slightly change its position in the membrane. The initial structures are manually positioned in the membrane without an exact position of z. Figure 8 shows the results of an interactive session of OmpA insertion. The session lasted less than 10 min, which is significantly shorter than the duration of the automatic sampling, which required about 1 h of calculation time. The results show similar plots to the automatic exploration.

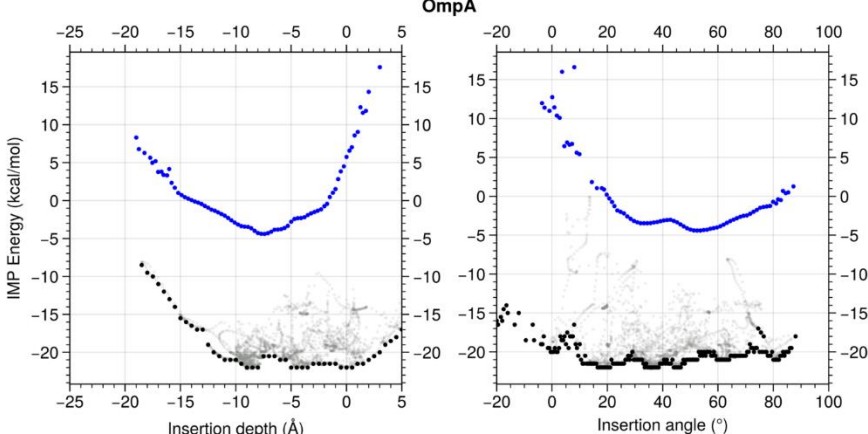

**Figure 8.** Interactive membrane insertion of OmpA. The blue dots correspond to the energy values of the Monte Carlo reference calculations. The windows for insertion depth (Å) and insertion angle (°) remain unchanged. The results of our own experiments are represented by the gray and black dots. The black dots are the minimum energy values at a given insertion angle or depth, rounded to the nearest multiple of 0.5. The gray dots correspond to all other energy values. Both plots have a common *Y*-axis on which the IMPALA energy term is shown in kcal/mol.

### 3.3. Graphical Analysis of Interactive Insertion Experiments

We designed a graphical representation to show the details of protein insertion. As shown in Figures 9 and 10, it is possible to determine the most favorable insertion angle and the associated roll for a given insertion depth. The diagrams show us in detail the region of most favorable insertion depth, with a region at $-10°$ corresponding to two categories of narrow insertion angles, $35°$ and $45°$, where we see an increase in atoms in the interfacial region of the membrane. The second region in the core of the membrane around 0 Å is related to the lower insertion angles around $15°$ to $20°$, where we suspect that the protein is more horizontal within the hydrophobic core, as the percentage of particles at the interface decreases in this region, as indicated by the green curve. Note that intuitively, the insertion angle does not give us a clear idea of the overall position of the structure, and the chosen definition of the insertion angle for β-barrel proteins such as OmpA does not pass through the barrel axis.

The energy barrier between $-10$ Å and $-5$ Å can be equated with a transition zone between the two regions mentioned above. This transition involves straightening of the protein reaching higher angles in the membrane, with partial exposure of the protein in the hydrophilic phase, as shown by the indentation in the orange curve. These results, together with the interactive simulations, give more credible information about the transitions between the favorable angular and depth zones than the absolute values, which must be related to uncertainties associated with the choice of atomic radii, rigid body dynamics, atomic transfer energies, IMPALA equations, etc. The core of this work lies in

the implementation of an interactive insertion protocol that allows the user to control and quickly simulate all input parameters. The interactive simulations of our protocol with the simulation–visualization coupling allow us to provide live visual feedback and to capture and analyze positions after the fact, as shown in Figure 11.

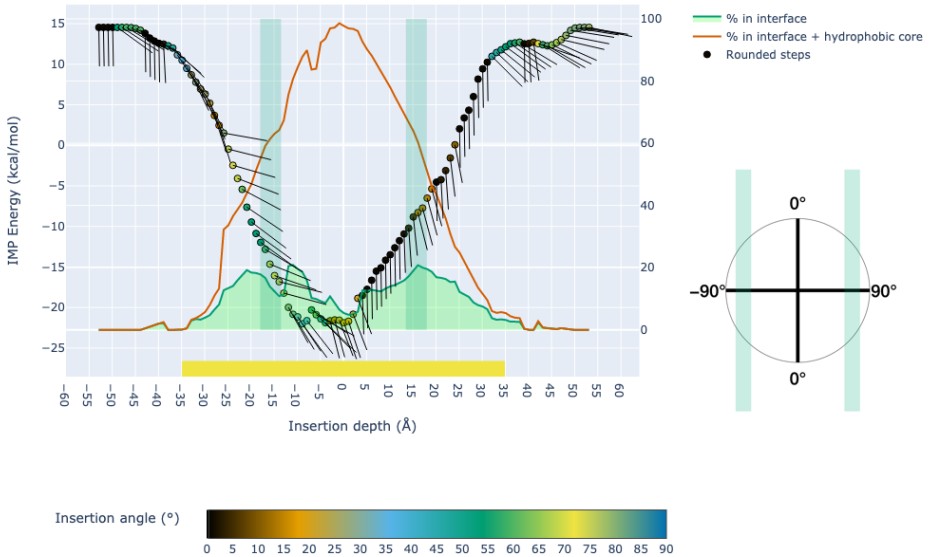

**Figure 9.** Detailed illustration of a membrane insertion experiment performed for OmpA. The colored circles mark the steps with the most favorable insertion angle at a given insertion depth. Each step can be associated with its IMPALA constraint (left vertical axis) corresponding to its insertion depth (horizontal axis), and the color provides information about the insertion angle; see color scale for correspondence. A black line starts at each step and its slope in the diagram represents the insertion angle, ranging from $0°$ (vertical line) to $90°$ (horizontal line). This is shown schematically in the reference circle on the right side of the figure. The vertical bars in transparent green represent the interface regions of the IMPALA membrane model. The green curve with the filled transparent-green region and the orange curve represent the percentage presence of atoms in the interfacial regions and in the whole membrane region, respectively. These two curves can be read on the right vertical axis. The yellow bar at the bottom of the graph indicates the geometric range of the OmpA protein, i.e., the maximum distance between its center of mass and the most distant atom, centered at a position of insertion depth 0.

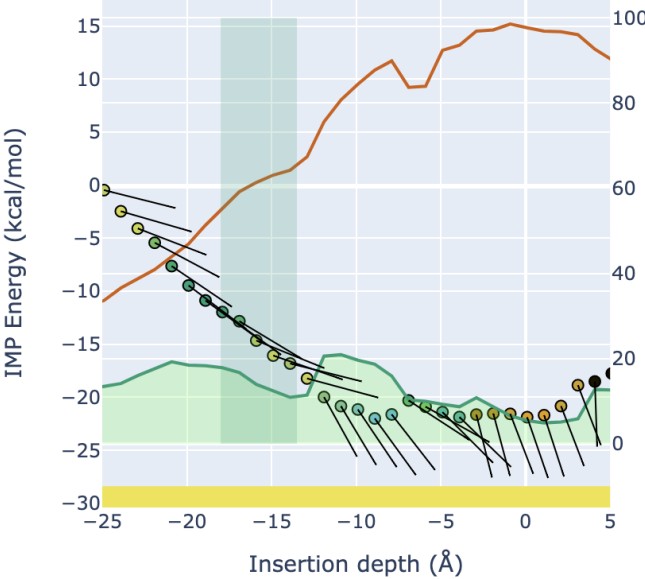

**Figure 10.** Enlargement of Figure 9, with focus on the area spanned by the reference insertion calculations.

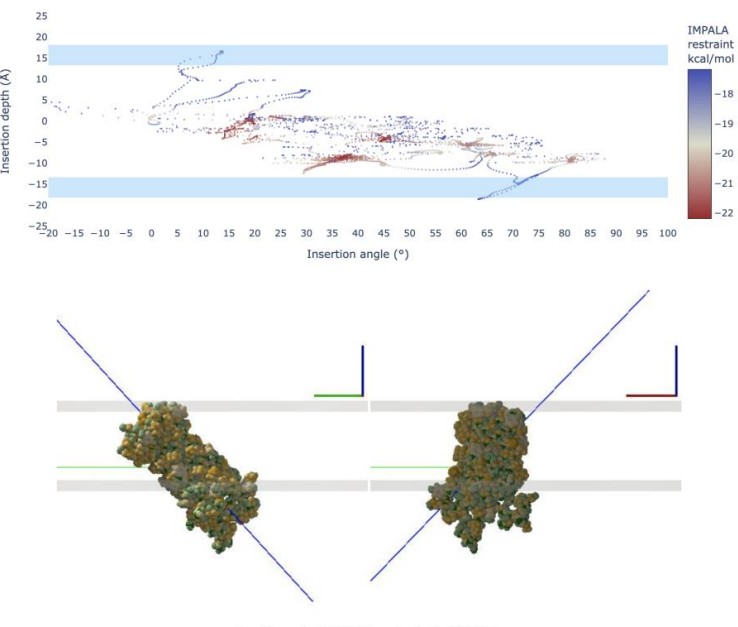

**Figure 11.** Interactive analysis showing the trajectory of the OmpA protein during insertion into the membrane. At the top, one can see the insertion depth versus the insertion angle, with the membrane shown as blue rectangles. The user can explore the scene captured by the UnityMol visualization software (see below) simultaneously in the *X* and *Y* axis viewing angles. The green horizontal line is a marker for the center of mass of the protein. The blue line is a line passing through the insertion vector. The current insertion depth and insertion angle are displayed.

The progress of an interactive experiment can be displayed live, as shown in the two supplementary videos, or from recorded data, as shown in Figure 11. When recording, the user can perform an interactive analysis with brushing and linking, where each combination of insertion depth and angle can be selected individually and thus visually represented by scene recordings previously made during the interactive session with UnityMol.

*3.4. Graphical Analysis of Exhaustive Scanning Experiments*

The results of an extensive automated scanning experiment can, of course, be displayed as a heatmap or density plot as depicted in Figure 12. The heatmap visualizes the energies in terms of insertion depth and angle, while the density plot shows the regions with the densest favorable energy zones. In particular, the two transition zones mentioned above are clearly visible in this diagram. In principle, such diagrams could also be derived from an interactive experiment if sampling is not too sparse.

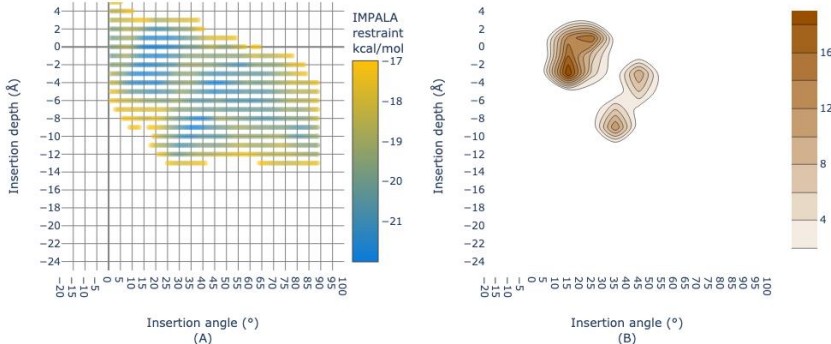

**Figure 12.** Heatmap and density plots of automated sampling of membrane insertion of OmpA. The range of energies considered is between the minimum and $-17$. (**A**) is a heatmap to visualize the energies in terms of insertion depth and insertion angle, and (**B**) is a density plot to show the regions with the densest favorable energy zones.

### 3.5. Comparison with Molecular Dynamics Data from Four Membrane Proteins

We examined a sample of four membrane proteins for which we had access to molecular dynamics simulations to further evaluate our implementation. Our dataset includes two beta-barrel membrane proteins, OmpT [34] and OmpX [35]; an ion-channel with an alpha-helical transmembrane domain, Glic [36]; and a G-protein coupled receptor, BLT2 [37]. Figure 13 shows the time series of insertion depth and insertion angle compared to the values predicted by our implementation of the IMPALA approach. The results highlight the intrinsic fluctuations characteristic of molecular dynamics simulations of membrane systems. In some cases, very good agreement is obtained, such as for the Glic or BLT2 insertion angles, while in other cases, such as for the Glic insertion depth, discrepancies are evident. In this particular case, this observation could be due to the thicker POPC membrane used in this simulation, e.g., compared to DMPC. The same argument may hold for the BLT2 insertion depth, as an even thicker PEA membrane was used. For the Omp simulations, it is obvious that at least OmpT is not fully stabilized with respect to membrane insertion, which generally suggests that simulations on the order of 10 s of nanoseconds are not sufficient to unambiguously assess membrane insertion. This is further confirmed by an alternative MD dataset extracted from the MemProtMD database [7] by analyzing the last snapshot deposited. Our IMPALA predictions are in good agreement with this data. For OmpX, our insertion depth is between the database value of +9.6 Å and the MD time series shown, which fluctuates around 0 Å, while there is good agreement for the insertion angle of both MD datasets, which is about 10 degrees higher than in our calculation. For Glic, the MemProtMD data are coherent and close to the IMPALA predictions. No reference data for BLT2 exist in MemProtMD.

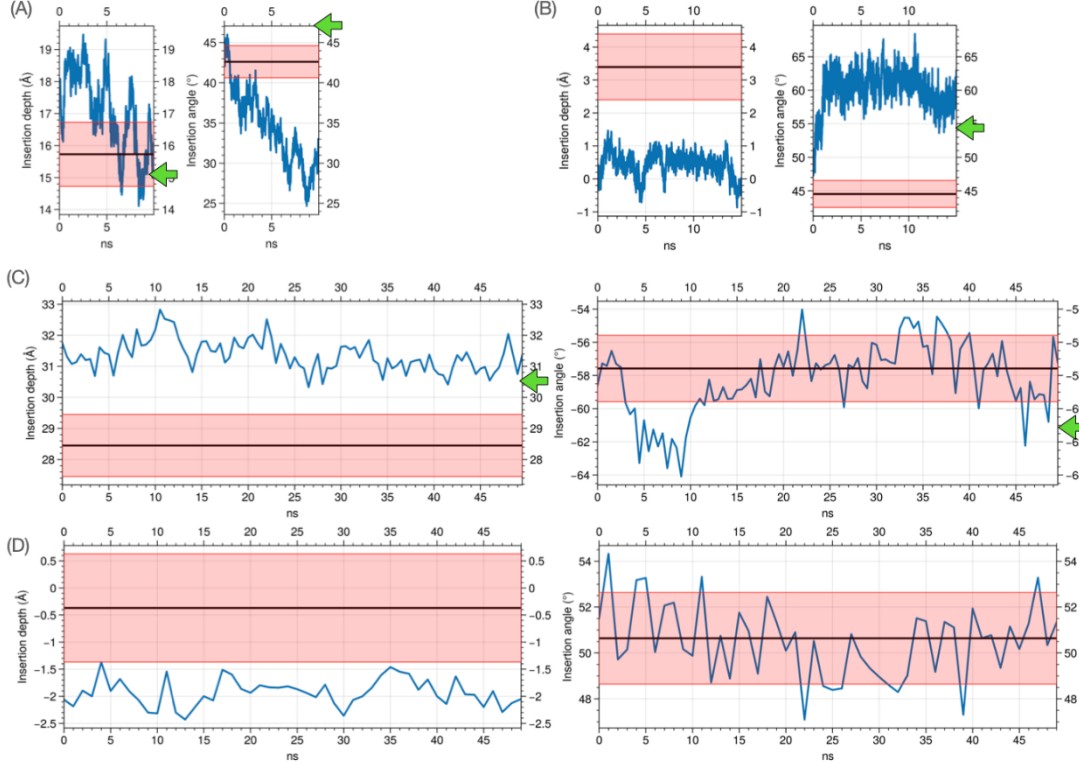

**Figure 13.** Insertion depths and angles for molecular dynamics simulations of (**A**) OmpT, (**B**) OmpX, (**C**) Glic, and (**D**) BLT2 with total lengths of 10, 15, 300, and 500 ns, respectively. For Glic and BLT2, only the last 50 ns of the simulation are shown. The optimized IMPALA insertion depth and angle are superposed as black line onto the molecular dynamics time series, with indication of an estimated minimal error of ±1 Å and ±2 degrees. We added green arrows to indicate values at the end of a set of alternative MD simulations deposited in the MemProtMD database [7] for comparison.

## 4. Discussion

The results obtained for the insertion of OmpA are largely consistent with the earlier results of Brasseur, thus confirming the general implementation of the IMPALA method. However, our comparison reveals a certain number of discrepancies that require a more detailed discussion. In particular, the energy profile of the insertion angle appears flattened compared to the original data. A fundamental problem is the fact that we do not have enough details to reproduce the experiment under 100% identical conditions, since these details were not included in the original publications, the actual code is not available, and there are contradictions between different publications by the same team. This is a common problem in computational research nowadays [38,39]. We had not anticipated the problem of reproducibility, otherwise we might have chosen a different method for calculating membrane insertion, but Brasseur's approach was appealing because of its simplicity and computational efficiency, which lends itself very well to interactive studies. The equations are simple and have been checked several times by different team members, so we are confident that the discrepancies are due to other sources. We describe the various sources of discrepancy with Brasseur's results that we have identified and discuss each case.

It should be noted that computational power has improved significantly over the past 24 years, and we expect our sampling to be more extensive than in the reference work. In the automatic scans, we performed 2.3 million energy evaluations, including insertion depth, insertion angle, and roll angle with steps of 1 Å spacing and 1 degree angle, while Brasseur's team typically performed 20-fold fewer calculations, with 0.1 million Monte Carlo tries, mentioning coarser moves with an insertion depth shift of $\pm 4$ Å random translation and a random rotation of $\pm 5$ degrees. Figure 7 shows that the IMPALA energy range calculated by Brasseur is also visited by our calculations. However, our new implementation finally stabilized systematically at even lower energies than in the reference work, consistent with a more comprehensive sampling. In addition, the earlier work highlighted the starting positions for each run, suggesting that the sampling may have been insufficient to correct for a poorly chosen starting position and hence intrinsically incomplete. As mentioned earlier, it is unclear whether or not the roll angle parameter was explicitly considered. It should also be noted that it is unclear whether the energy has been scaled or shifted for convenience in the original calculations; in particular, no energy units are reported in any of the published graphs on insertion calculations.

Another key element is that the calculations depend on a number of parameters and choices that are not explicitly explained in the reference papers. While we have attempted to reconstruct the original conditions as best we can from the information available to us, in the absence of explicit information, this is obviously limited. However, apart from the possibility that the exact parameters are wrong, there is no reason to expect intrinsic discrepancies between Brasseur's implementation and our approach. In summary, the uncertainties relate to parameters such as the atomic radii, the empirical factor $a_{lip}$, the addition of hydrogen atoms, the calculation of SASA, the exact equations for the IMPALA terms, and the sampling conditions of the Monte Carlo runs. We discuss these elements and our attempts to clarify the optimal parameter choices in more detail in Appendix A. We also provide a second example on a simpler system, Melittin, where good agreement is reached particularly for the insertion angle exploration.

As we have shown, a direct comparison with the reference data of the IMPALA method is limited by a number of uncertainties. As an additional evaluation of our implementation for membrane protein positioning, we performed a comparison with molecular dynamics data for four membrane proteins, OmpT, OmpX, Glic, and BLT2, which have beta-barrel and alpha-helical architectures of different sizes. This comparison shows that our predicted insertion depths and angles are generally consistent with observations in these detailed all-atom simulations. Nevertheless, discrepancies can occur due to a number of factors, such as intrinsic fluctuations and slow convergence in the simulations or insertion in bilayers with different thicknesses compared to the implicit model.

Both the interactive experiment and the automatic scanning of the parameter space provide comparable results, but the interactive approach provides a time gain of almost an order of magnitude. In addition, live observation of the trajectory of the protein in response to the manipulation provides intuitive information about the specific properties of the insertion. Positive informal feedback from several test users of our implementation confirms our working hypothesis that real-time interactive positioning of proteins in model membranes provides immediate insight into the insertion process that closely resembles the results of more extensive calculations.

In our opinion, it is important to clearly separate the inherent advantages of an interactive approach to membrane protein positioning from the potentially limited accuracy and precision that can be expected from the simple models we have implemented. The precise computational algorithm can be easily replaced if a more accurate computational method is found. Even with its current limitations in distinguishing insertion angles due to the relatively flat energy profile, which is not very discriminating, the approach is particularly promising for integrative modeling. In this context, the interactive simulation is guided by external data, for example, from EPR or NMR experiments. The integrative modeling data usually ideally complement the interactive approach by providing experimental clues to correct any inaccuracies and heighten the flat energy profile. Insertion depths relative to the membrane center can be derived from EPR experiments and provide additional constraints along the lines of the study described in [8]. In an experiment with an early prototype of our implementation (an even coarser model, using only single beads for each amino acid), a colleague could combine the interactive experiment with NMR data to resolve the position of a peptide in a membrane environment [40].

In the future, we would like to study a broader range of membrane proteins, explore the performance limits for large systems, and extend the methodology to test flexible models and more complex processes such as membrane aggregation or larger, multi-membrane spanning assemblies.

**Supplementary Materials:** The following supporting information can be accessed online: Video S1: UMol preview OmpA rigidbody at https://youtu.be/RRk2-8humrM, Video S2: UMol preview OmpA springs at https://youtu.be/r5uR9aOG044.

**Author Contributions:** Conceptualization, N.F. and M.B.; methodology, A.L., N.F. and M.B.; software, A.L., B.L., H.S. and N.F.; validation, A.L., N.F. and M.B.; investigation, A.L., N.F. and M.B.; resources, A.L., B.L. and H.S.; data curation, A.L. and N.F.; writing—original draft preparation, A.L. and M.B.; writing—review and editing, all authors; visualization, A.L.; supervision, M.B.; project administration, M.B.; funding acquisition, M.B. All authors have read and agreed to the published version of the manuscript.

**Funding:** This research was funded by the "Initiative d'Excellence" program from the French State, grants "DYNAMO", ANR-11-LABX-0011, and "CACSICE", ANR-11-EQPX-0008). M.B. thanks Sesame Ile-de-France for co-funding the display wall used for data analysis, grant n° 13016382.

**Institutional Review Board Statement:** Not applicable.

**Informed Consent Statement:** Not applicable.

**Data Availability Statement:** The software packages used for our implementation, UnityMol, MD-Driver, and BioSpring, are open source projects. The extensions to these software packages presented in this study are immediately available on request from the corresponding author and will become publicly available after further testing by including them in the next software package release. To help experiment with our approach, we provide ready-to-use compiled versions for selected platforms along with specific input data. These data are openly available at https://recherche.data.gouv.fr/, accessed on 1 November 2022, at doi 10.57745/NSHIWZ.

**Acknowledgments:** We thank Alex Tek for the preliminary work he did for his dissertation, which formed the basis for our study. We also thank Xavier Martinez for UnityMol software support.

**Conflicts of Interest:** The authors declare no conflict of interest. The funders had no role in the design of the study; in the collection, analyses, or interpretation of data; in the writing of the manuscript; or in the decision to publish the results.

## Appendix A

Here, we explain the uncertainties that exist in our implementation of the IMPALA method due to missing information from the literature, and how we attempted to determine the most likely parameter combination to reproduce the original calculations. One of the causes of the discrepancy is in the actual sampling. In addition to our exhaustive parameter grid search, we implemented an experimental simple Monte Carlo approach to mimic Brasseur's sampling approach. However, insufficient details were reported on the exact implementation of the Monte Carlo moves, so we had to make assumptions about the translation and rotation conditions. The reference work mentions rotation, but it is unclear whether only a single angle was considered or whether both the insertion angle and the roll angle were sampled separately. Our test results mainly confirmed that the sampling approach had an impact on the results and showed significant variability between 10 runs, while Brasseur only performed two replicates.

The equations used in the early publications of the IMPALA method vary without any justification. In particular, the definition of **C(z)** in one publication was given as water content at a certain level and ranged from 0 to 1, whereas in another publication, it was shifted to the range $-0.5$ to $+0.5$. We tested both definitions, which are related by a simple shift in the resulting energies, but ultimately used the latter definition. The reason is that the alternative implementation did not make sense from a physicochemical point of view, as it implied that the positioning of the protein always resulted in an energy penalty and never in a favorable energetic contribution. A similar problem was observed for the $a_{lip}$ parameter, which was reported with a positive or negative sign depending on the publication, and its units are never explicitly indicated. The addition of hydrogens is not explicitly mentioned: in early work on peptides, hydrogens are clearly present. In later work on membrane proteins, the underlying PDB structures do not contain hydrogen atoms, and it is not known whether or how the hydrogen atoms were eventually added. If they had been omitted, the calibration of the per-atom transfer energy contributions would need to be adjusted, but this has not been reported. It seems likely, therefore, that the hydrogen atoms were added. If so, the parameters chosen for the addition of hydrogen could affect the results. We have already mentioned the exact atomic radii as important parameters for the calculation and took these parameters from one of the original articles. These parameters are important for the calculation of SASA, which in turn depends on the exact algorithm and parameters such as resolution, which we discuss in more detail in Section 2.3.3.

A final element concerns the measurement of insertion depth and angle. The depth, measured as the position of the center of mass, leaves little doubt. However, as far as the definition and measurement of the angle are concerned, as already mentioned in the manuscript, we found that there are incoherences between text and illustrations in the original works on OmpA, which means that we are not 100% sure to measure exactly the same angle as it is given in the literature.

In addition to the experiments with OmpA, a complete protein, we attempted to reproduce previous work with the simpler peptide melittin described in [18]. An alpha-helical model was used and the hydrogens are clearly present. To reproduce this experience, we used our experimental Monte Carlo implementation. Figure A1 shows the comparison between the originally reported results and our experiments for the insertion angle. It can be seen that several important features of the graph are well reproduced, particularly three minima and a saddle point. The good agreement between our experiment and the reference implementation emphasizes that the results are well reproduced if the uncertainties can be removed.

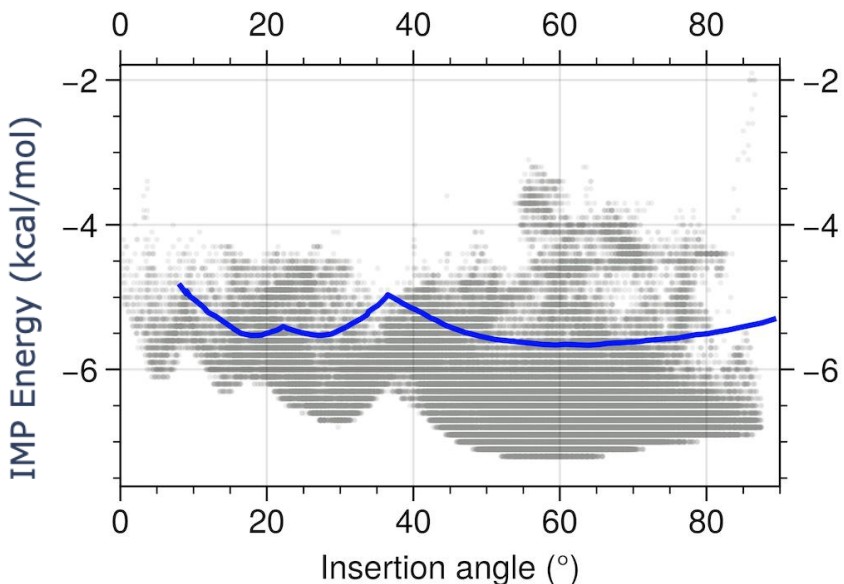

**Figure A1.** Profiles of the simulation of melittin. In the graph, a dot corresponds to a Monte Carlo step representing the sum of all IMPALA restraints against the insertion angle. The gray dots correspond to sampling in our implementation; in blue is the minimum energy profile of Ducarme et al. for comparison.

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
