# Peer review of "Fast and Interactive Positioning of Proteins within Membranes"

_algorithms, doi:10.3390/a15110415_

Round 1

Reviewer 1 Report

In this work, the authors proposed an interactive approach to place proteins within membranes, which could be useful for the field and even for education. I generally support the publication of the manuscript if the following concerns being properly addressed.

1.     There are non-negligible differences in insertion angles (Figure 7) between the approach proposed by the authors and the reference Monte Carlo calculations. The authors may have more detailed discussions and propose solutions.

2.     The authors mentioned the discrepancies between their predictions and experiments could be ascribed to the case that their approach didn’t capture enough details to reproduce the experiment under 100% identical conditions. It will be essential to compare a couple of model membrane systems (comparable to the setup of their approach) with membrane proteins using all-atom/Martini coarse-grained MD simulations.

Author Response

We thank the reviewer for the constructive criticism and for pointing out important points that helped us to improve the manuscript and provide more relevant details. We have addressed both points as follows   Regarding the differences in the insertion angles, we first noticed that the digitized data that were overlaid were in the wrong units (the values given were kJ-mol, whereas the graph contained kcal-mol units). After the corrections, the reference data in Figure 7 shows a flatter profile that is more consistent with our calculations. We also now provide much more detail and discussion. As suggested in the journal instructions and submission template, we have moved the additional detailed explanations that would disrupt the flow of the main text, but are nonetheless important to understand and follow the research results presented at the end of the article, to the appendices. To summarize our main findings: When the number of sources of discrepancies and uncertainties is reduced, we achieve very good agreement. To illustrate this, we now show data for the simpler peptide melittin, for which we also implemented a Monte Carlo sampling approach to mimic the original calculations as closely as possible. As now shown in Figure A1, we obtain very good agreement for the angular profile by reproducing three marked minima and one saddle point. We believe that with the additional explanations, we have addressed all the concerns of the reviewer on this point.   Regarding the second point, we directly followed the referee's suggestion and compared a number of model membrane systems with membrane proteins using all-atom MD simulations. We selected four membrane proteins for which we had access to molecular dynamics simulations, including two beta-barrel membrane proteins, an ion channel with an alpha-helical transmembrane domain, and a G-protein-coupled receptor. There is agreement between the molecular dynamics sampling and the predicted insertion depths and angles in some cases, and discrepancies in others that may be explained by insufficient sampling or convergence, bilayers that are thicker than the model assumed by Brasseur etc.. Overall, we believe that the comparison of MD provides the reader with interesting data to better understand the limitations and potential of the current model. For example, it reproduces the insertion angle surprisingly well for both longer MD simulations.   It should be noted that considering eventual discrepancies and inaccuracies, those can be eliminated if the interactive approach is used, for example, in the context of integrative modeling where additional data, e.g., from EPR or NMR experiments, are used to guide and constrain the simulation. The experimental cues allow inaccuracies to be corrected and the energy profile to be sharpened. In an experiment using a very early and even rougher prototype of our implementation, a colleague was actually able to combine the interactive experiment with NMR data to determine the position of a peptide in a membrane environment.

Reviewer 2 Report

The article presents a potentially very useful algorithm that has been implemented and verified in a technically and scientifically sound way. I am sure many people in the field will appreciate it and highly recommend its publication.

Author Response

We thank the reviewer for the positive feedback. Following the reviewer's assessment that a minor spell check was needed, we reviewed the English language, typography, and grammar and corrected 21 errors. We also improved the clarity and wording of the text.

Round 2

Reviewer 1 Report

The authors have properly replied to my concerns and revised the manuscript accordingly. Hence, I suggest the publication of the manuscript in the current form.